DATA RELEASE

# *De novo* construction of a transcriptome for the stink bug crop pest *Chinavia impicticornis* during late development

Bruno C. Genevcius[1,*] and Tatiana T. Torres[1]

1 Department of Genetics and Evolutionary Biology, University of São Paulo, São Paulo, SP, Brazil

## ABSTRACT

*Chinavia impicticornis* is a neotropical stink bug of economic importance for various crops. Little is known about the development of the species, or the genetic mechanisms that may favor the establishment of populations in cultivated plants. Here, we conduct the first large-scale molecular study of *C. impicticornis*. Using tissues derived from the genitalia and the rest of the body for two immature stages of both males and females, we generated RNA-seq data, then assembled and functionally annotated a transcriptome. The *de novo*-assembled transcriptome contained around 400,000 contigs, with an average length of 688 bp. After pruning duplicated sequences and conducting a functional annotation, the final annotated transcriptome comprised 39,478 transcripts, of which 12,665 were assigned to Gene Ontology (GO) terms. These novel datasets will be invaluable for the discovery of molecular processes related to morphogenesis and immature biology. We hope to contribute to the growing body of research on stink bug evolution and development, as well as to the development of biorational pest management solutions.

**Subjects** Genetics and Genomics, Agriculture, Functional Genomics

**Submitted:** 22 October 2020

* Corresponding author. E-mail: bgenevcius@gmail.com

Preprint submitted at https://doi.org/10.1101/2020.12.04.412270

## INTRODUCTION

The green stink bug *Chinavia impicticornis* (Hemiptera, Pentatomidae; Figure 1) is a neotropical species with wide distribution across South America. The species is an economically important polyphagous pest, which has been reported to feed on 14 plant species [1]. Most of the damage it causes is in soybean and closely related crops. However, the stink bug's ability to switch to less preferred hosts in anomalous conditions may help to explain how populations have successfully established in different crops [2, 3].

As well as *C. impicticornis*, many other stink bug species are responsible for hundreds of millions of dollars-worth of agricultural damage every year across the world. For example, in 2011, the brown marmorated stink bug (*Halyomorpha halys*) was responsible for $37 million in losses to apple growers in the USA [4]. Growing applied and basic research effort towards mitigating this damage has certainly contributed to our knowledge of the biological aspects and management strategies for stink bug pests [5]. Most efforts have focused on sampling methods, taxonomic identification, insecticide effectiveness, and population monitoring [6, 7]. However, traditional pest control techniques have the potential to inflict serious ecological disturbances, and to increase selection for resistant crop pest lineages. The development of biorational solutions largely depends on detailed

**Figure 1.** Picture of the crop pest the green stink bug *Chinavia impicticornis*. Picture courtesy of Marcio Cezario.

comprehension of the underlying biology of these pests [8]. In this context, 'omics studies offer promising species-specific and environmentally friendly tools [9]. More specifically, transcriptomic approaches provide the foundation for identifying gene targets associated with pheromones, pesticide resistance, and other features with potential for pest management. Conversely, the transcriptomes of only five species of stink bug have been sequenced to date: the brown marmorated stink bug (*Halyomorpha halys* (Stål)) [10], the harlequin bug (*Murgantia histrionica*) [9], the southern green stink bug (*Nezara viridula*) [11], the brown stink bug (*Euschistus heros*) [12], and the predatory stink bug *Arma chinensis* [13]. Genomic data are even more scarce, comprising four genomes assembled to date: the the brown marmorated stink bug (*Halyomorpha halys* (Stål), [14]), the brown stink bug (*Euschistus heros*, NCBI bioproject PRJNA489772), redbanded stink bug (*Piezodorus guildinii*, PRJNA263369), and the anchor stink bug (*Stiretrus anchorago*, PRJNA345234).

Here, we target this gap by documenting and characterizing the first transcriptome for the neotropical stink bug *Chinavia impicticornis* (Hemiptera, Pentatomidae). Our study uses tissue from nymphal stages for two reasons. First, because management strategies are known to affect nymphs and adults in different ways, and nymph transcriptomes are extremely scarce for stink bugs (*e.g.* [10]). Second, developmental transcriptomes provide invaluable data for the discovery of molecular processes underlying morphogenesis. We focused on the fifth instar because this is the stage where morphological sexual differentiation takes place. These resources may be helpful for the growing body of evolution and development research being carried out on the green stink bug [15, 16].



## METHODS

### Samples

Our colony of *Chinavia impicticornis* was fed on green bean pods (*Phaseolus vulgaris*) and reared under the following conditions: 26 ± 1 °C, 65 ± 10% relative humidity (RH), and a photoperiod of 14 h light:10 h dark. We isolated RNA from immature male and female insects at two developmental stages: the beginning and the end of the fifth nymphal instar (2 h, and 7 days after molting from fourth to fifth instar, respectively). The fifth nymphal instar in our controlled conditions taking an average of eight days. We included three individuals per sex per stage, amounting to 12 specimens. RNA was extracted from genital tissue, and body tissues to construct separate libraries for these tissue types. Genitalia are important for both species delimitation and sex identification in true bugs. Therefore, with this approach, we expect to generate data that will be particularly useful for studying the mechanisms of sex determination and speciation. For RNA sequencing, each genital sample was sequenced separately (*n* = 3 genitalia per sex per sample), while bodies of the same sex and developmental stage were pooled to generate a single library (*n* = 1 body per sex per stage).

### RNA extraction and sequencing

Frozen specimens (−80 °C) were sexed and had their genitalia separated under a stereoscope. RNA extraction of genitalia and bodies was conducted using Trizol reagent, following the manufacturer's protocol (Invitrogen, Life Technologies). RNA was quantified with Qubit 2.0 (Thermo Fisher Scientific, USA) and then sent for sequencing at the Centro de Genômica Funcional (ESALQ-USP) using an Illumina HiSeq 2500 sequencer (RRID:SCR_016383). Samples were pair-end sequenced with 100 bp and a target depth of 20 million read pairs per sample.

### Transcriptome assembly and annotation

Redundant sequences were removed using a custom Perl script to decrease computational usage for transcriptome assembly [17, 18]. *De novo* assembly was conducted in Trinity v.2.4.0 (RRID:SCR_013048) [19] with concatenated samples, a minimum contig length of 199, and other parameters set to default values.

The lack of reference genome meant that transcriptome annotation was conducted using FunctionAnnotator [20]. This is a web-based tool that blasts sequences against the National Center for Biotechnology Information (NCBI)'s non-redundant (NR) protein database. FunctionAnnotator was also used for functional characterization, employing the B2G4PIPE engine to assign Gene Ontology (GO) terms. To report the functional characterization of the transcriptome, only contigs matching to arthropod species were kept.

### Quality control (QC)

The quality of raw reads was assessed using the program FastQC v.0.11.9 (RRID:SCR_014583) [21]. A quality control plot was constructed using MultiQC v.1.9 [22] and is represented in Figure 2. Raw reads of all samples had good quality. Low quality sequences and adapters were removed using Trimmomatic v. 0.36 (RRID:SCR_011848) [23] with the following parameters: LEADING:3 TRAILING:3 SLIDINGWINDOW:4:20 MINLEN:36. An average of 11.5 ± 0.006% of raw reads was removed during the trimming process.

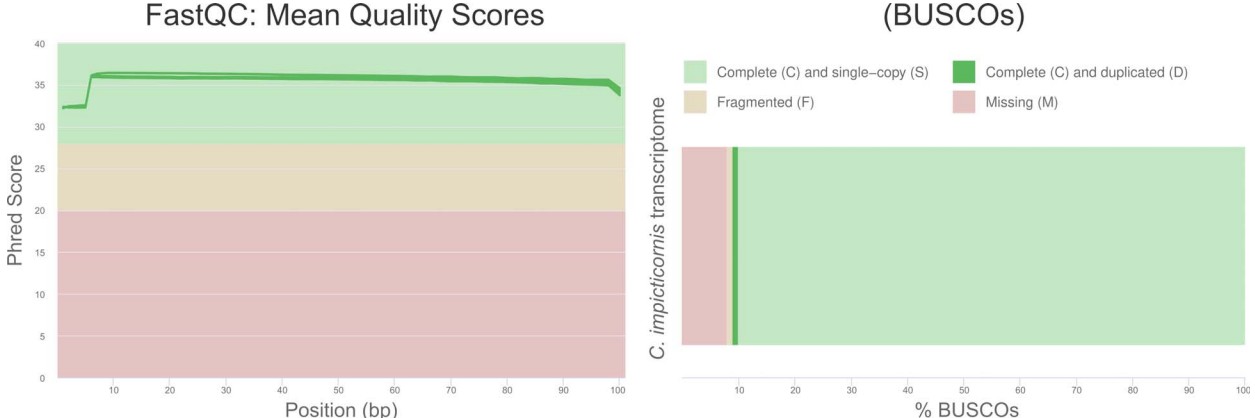

**Figure 2.** Quality control of raw reads using FastQC (left) and analysis of transcriptome completeness using BUSCO (right).

**Table 1.** Descriptive statistics of raw RNA-seq data and transcriptome assembly of the stink bug *Chinavia impicticornis* compared with all other transcriptomic studies of pentatomids.

|  | *Chinavia impicticornis* | *Arma chinensis* | *Euschistus heros* | *Halyomorpha halys* | *Murgantia histrionica* | *Nezara viridula* |
|---|---|---|---|---|---|---|
| Platform | Illumina HiSeq. 2500 | Illumina HiSeq. 2000 | Illumina HiSeq. 2000 | Illumina HiSeq. 1000 | Illumina HiSeq. 2000 | Illumina HiSeq. 2000 |
| Raw reads | 522,644,434 | nr | 126,455,838 | 439,615,225 | nr | 284,400,000 |
| Contigs assembled | 392,739 | 112,029 | 83,114 | 248,569 | 526,403 | 299,148 |
| GC content (%) | 35.04 | nr | 37.12 | nr | nr | nr |
| Mean contig length (bp) | 688 | 250 | 1,000 | nr | 812 | nr |
| Pipeline | Trinity | Trinity | Trinity | Trinity | Trinity | Trinity |
| GOs annotated | 5,087 | nr | 143,806 | 1,099 | 1,348 | 771 |
| Reference | This study | [13] | [12] | [10] | [9] | [11] |
| BioProject accession | PRJNA666218 | PRJNA180995 | PRJNA488833 | PRJNA242849 | PRJNA302154 | PRJNA472074 |

Abbreviations: bp, base pair; GO, Gene Ontology; nr, not reported.

To avoid redundant transcripts in the transcriptome, the shortest isoforms were removed using a built-in Trinity script. Transcriptome completeness was then assessed using Benchmarking Universal Single-Copy Orthologs (BUSCO) v.4.1.4 (RRID:SCR_015008) [24] against the Hemiptera ortholog database with default parameters. BUSCO analysis revealed appropriate levels of completeness for the assembled transcriptome (Figure 2).

## TRANSCRIPTOME CHARACTERIZATION

RNA sequencing generated approximately 5 million raw reads (Table 1). The *de novo*-assembled transcriptome contained almost 400,000 contigs, with a GC content of 35.04% and a mean contig length of 688 bp. After removing the shortest isoforms, 268,000 contigs remained. Of these, FunctionAnnotator successfully matched 39,478 contigs in the NCBI database, of which 36,329 aligned with metazoans and 33,871 with arthropods. Most contigs were annotated against Hemiptera sequences; more specifically, against the genome of *Halyomorpha halys* (Figure 3). All metrics had comparable values among previous studies with stink bugs (Table 1).

For the 33,871 transcripts that aligned with arthropods, functional annotation resulted in 12,665 contigs with GO terms assigned, amounting to 5087 GO terms across the three categories: biological processes, molecular functions, and cellular components. For the

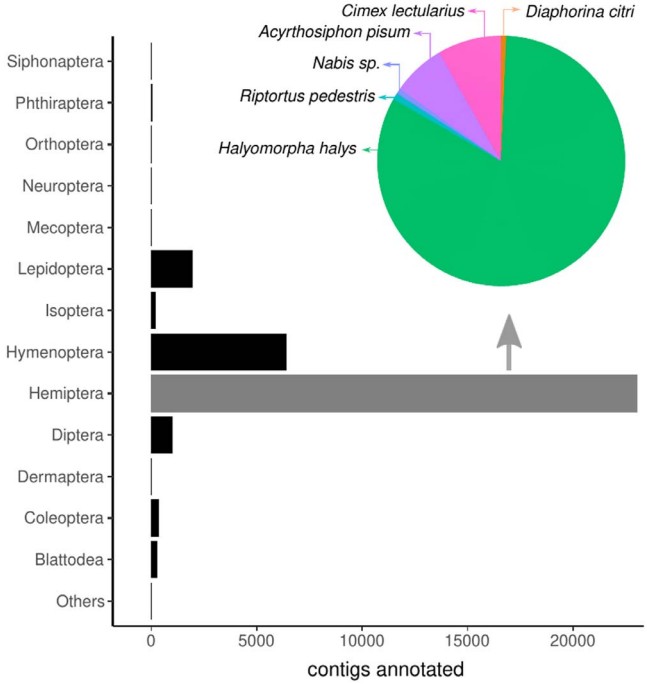

**Figure 3.** Diversity of insect orders from which the contigs of the *Chinavia impicticornis* transcriptome were annotated.

function 'biological process', the most commonly represented GO term among annotated transcripts was 'RNA-dependent DNA biosynthetic process' (Figure 4). For 'molecular functions', the most common GO term was 'RNA-binding', while in 'cellular components', the most represented term was 'nucleus' (Figure 4).

## RE-USE POTENTIAL

Genomic and transcriptomic resources are paramount for advances in pest control and for the understanding of the origins of morphological innovations. Despite the high species diversity and the economic importance of stink bugs, large-scale comparative molecular studies are still scarce for this group of insects. Here, we present the first large-scale molecular study with the Neotropical pest *C. impicticornis*. From an economic perspective, we encourage further use of our data to determine the genetic bases of adaptation to new hosts and insecticide resistance. In particular, comparative transcriptomic studies may help explain why some stink bug species are more or less adaptable to different crops in comparison to *C. impicticornis*. From an evolutionary perspective, our data may be useful to investigate both the mechanisms of species separation and sexual differentiation. As genitalia tend to be the most complex and rapidly evolving traits in insects, studying the genetic/developmental architecture of these structures may help understand the emergence of morphological barriers during speciation. Our study is the first to make available separate transcriptomes of male and female immature stink bugs. We also provide for the first time, transcriptome data of two different time-points of a single immature stage, which will potentially inform about developmental heterochrony on a finer scale. Lastly, we also hope to contribute useful data for phylogenomic studies of hemipterans.

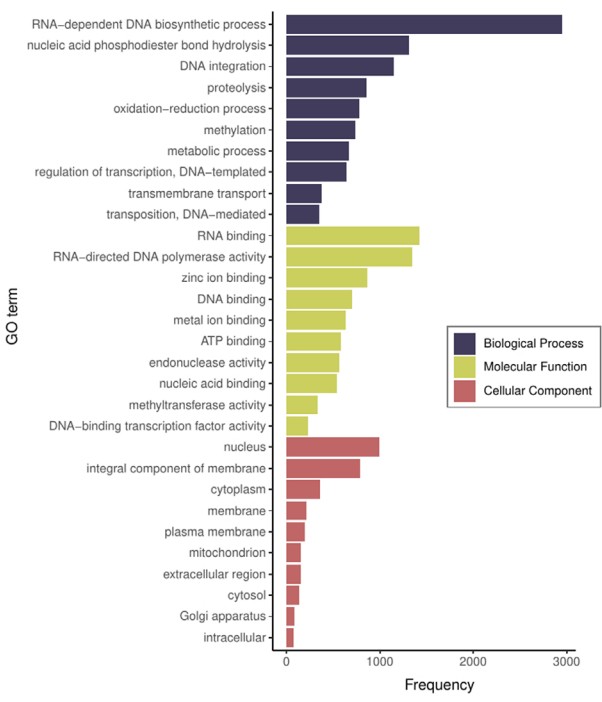

**Figure 4.** Top ten Gene Ontology (GO) terms for each category in the transcriptome of *Chinavia impicticornis*, as annotated by FunctionAnnotator.

## AVAILABILITY OF SUPPORTING DATA

Raw data are associated with the NCBI BioProject PRJNA666218 and are deposited in the Sequence Read Archive (SRA) under the accession codes: SRR12762704, SRR12762703, SRR12762702, SRR12762696, SRR12762692, SRR12762689, SRR12762690, SRR12762691, SRR12762693, SRR12762694, SRR12762695, SRR12762697, SRR12762698, SRR12762699, SRR12762700, and SRR12762701. The transcriptome assembly is deposited in the NCBI Transcriptome Shotgun Assembly Sequence Database (TSA) under the accession code GIVF00000000. Annotation, QC files and custom scripts are also available from the *GigaScience* GigaDB repository [18].

## ACKNOWLEDGEMENTS

We thank Gisele Antoniazzi and Denis Calio for their help to establish and maintain the insect colony, as well as Luiza Saad and Federico Brown for their support with RNA extraction. We also thank the reviewers Peter Thrope and Guillem Ylla and the Editors for their constructive criticism during the review. BCG was supported by Fundação de Amparo à Pesquisa do Estado de São Paulo (FAPESP) with a postdoctoral fellowship (proc. n. 18/18184-4). This work was supported by grants to TTT from FAPESP (grant 2016/09659-3).

## AUTHOR CONTRIBUTIONS

BCG handled the insect colony, conducted the experiments and RNA extraction. BCG and TTT analyzed the data and wrote the manuscript. All authors read and approved the final version of the manuscript.

## COMPETING INTERESTS

The authors have no competing interests to declare.

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
