## [Reviewer Report]

Reviewer name and names of any other individual's who aided in reviewer Peter ThorpeDo you understand and agree to our policy of having open and named reviews, and having your review included with the published papers. (If no, please inform the editor that you cannot review this manuscript.)YesIs the language of sufficient quality?YesPlease add additional comments on language quality to clarify if needed
Are all data available and do they match the descriptions in the paper? NoAdditional CommentsThey are submitted but still private. These need to be released. Are the data and metadata consistent with relevant minimum information or reporting standards? See GigaDB checklists for examples <a href="http://gigadb.org/site/guide" target="_blank">http://gigadb.org/site/guide</a>YesAdditional CommentsIs the data acquisition clear, complete and methodologically sound?YesAdditional CommentsIs there sufficient detail in the methods and data-processing steps to allow reproduction?YesAdditional CommentsIs there sufficient data validation and statistical analyses of data quality? YesAdditional CommentsIs the validation suitable for this type of data?YesAdditional CommentsIs there sufficient information for others to reuse this dataset or integrate it with other data?YesAdditional CommentsAny Additional Overall Comments to the AuthorSRA datasets need to be released. RecommendationMinor Revision

---

## [Reviewer Report]

Reviewer name and names of any other individual's who aided in reviewer Guillem YllaDo you understand and agree to our policy of having open and named reviews, and having your review included with the published papers. (If no, please inform the editor that you cannot review this manuscript.)YesIs the language of sufficient quality?YesPlease add additional comments on language quality to clarify if needed
While the text is mostly clear, I detected a few spelling mistakes (listed below) and there might be more that escaped my attention. I would recommend the authors to exhaustively check the MS. Line 53: “Stink bug” missing “bug”. Lines 39,58,69, and figures: Mixed usage of “Chinavia impicticornis” and “C. impicticornis”. After first appearance of the full name, authors should be consistent whether they keep using the full name or the abbreviation, but not mixing both.Are all data available and do they match the descriptions in the paper? NoAdditional CommentsThe authors report multiple accession numbers from NCBI including a BioProject ID. But they are not open and I was unable to check if the data match the paper descriptions. The TSA accession seems that has not yet been created and the MS displays a placeholder (GIVF00000000) in its place.Are the data and metadata consistent with relevant minimum information or reporting standards? See GigaDB checklists for examples <a href="http://gigadb.org/site/guide" target="_blank">http://gigadb.org/site/guide</a>NoAdditional CommentsMissing items from the checklist. 1) "Any perl/python scripts created for analysis process ". In Line 94 “using a custom Perl script [16]”, the authors provide citation but not the code. 2) "Full (not summary) BUSCO results output files (text) ". Is the data acquisition clear, complete and methodologically sound?YesAdditional CommentsThe end of the fifth nymphal instar dataset was obtained at “seven days after molting from fourth to fifth instar”. Could authors specify how many days is the 5th nymphal instar to have a better idea of how much longer is the 5th nymphal stage.

Could the authors briefly describe the rationale o behind choosing 5th nymphal and instead of other nymphal stages? They explain why nymphal stages were used instead of adults, but not why the 5th nymphal instar. Is there sufficient detail in the methods and data-processing steps to allow reproduction?NoAdditional CommentsI would appreciate if the authors could share the code/commands for removing redundant reads and performing the assembly as supplementary materials or in GitHub (recommended).


In the abstract, the authors describe 38,478 transcripts of which 12,665 had GO terms assigned. Is not clear where this number comes from. In line 120 is mentioned that “ 39,478 had successful matches in the NCBI”. Is there a type one of these two numbers (38,478 vs 39,478)? However, the MS says “we only kept contigs that matched to Arthropod species”, and this number is reported to be 33,871. I urge the authors to better explain the steps they followed and clarify where all these numbers come from.

Is there sufficient data validation and statistical analyses of data quality? YesAdditional CommentsUsing the whole insect body often includes contaminant RNAs from the gut microbiome, endosymbionts, viruses, and other microbiological specimens from the cuticles and environment. Since the authors do not filter out reads from possible contaminants before the assembly, I would appreciate it if they could perform a BUSCO analysis using the prokaryote database before and after the selection based on similarity to databases. This would allow estimating the number of contaminants in the original assembly and if they had successfully discarded after the selection.


Lines 126-127 are not clear. There are 12,665 contigs that have 5,087 GO terms. I deduce that there are 12,665 contigs that have at least 1 GO term, and that they contain 5,087 distinct GO terms. Could authors make it more clear on the text?Is the validation suitable for this type of data?YesAdditional CommentsIs there sufficient information for others to reuse this dataset or integrate it with other data?YesAdditional CommentsI don’t think that a dataset consisting of 2-time points (early and late) of the same sarge (nymph 5) can be considered a “developmental transcriptome”. I would urge the authors to change the terminology and title. 

In the abstract, the authors claim that this is the “ first genome-scale study with”. Since the study is only transcriptomic, I find it misleading to define it as “genome-scale study”.

1- I don’t think that a datasets consisting of 2 time points (early and late) of the same sarge (nymph 5) can be considered a “developmental transcriptome”. I would urge the authors to change the terminology and title. 

2- In the abstract, the authors claim that this is the “ first genome-scale study with”. Since the study is only transcriptomic, I find misleading to define it as “genome-scale study”.

3- In table 1 and line 117 the authors claim that they generated the highest amount of RNA-seq reads for pentatomids to date. However, for the Halyomorpha halys there are multiple available RNA-seq datasets not mentioned, which taken together I suspect that they would accede the data generated for C. Impicticornis. I would suggest to reduce the tone of this statement of L117. 

4- Additionally, there are at least 3 available genomes for pentatomidaes species. I think that this information should at least be mentioned in the introduction. 

5- In line 61, could the authors define “almost nonexistent”, how many are there?

Additionally, there are at least 3 available genomes for pentatomidaes species. I think that this information should at least be mentioned in the introduction. 

In line 61, could the authors define “almost nonexistent”, how many are there?Any Additional Overall Comments to the AuthorRecommendationMinor Revision